# Contrastive Fingerprinting: a Novel Website Fingerprinting Attack over Few-shot Traces

## ABSTRACT

Website Fingerprinting (WF) attacks enable passive adversaries to identify the website a user visits over encrypted or anonymized network connections. WF attacks based on deep learning have achieved high accuracy in identifying websites based on abundant training traffic traces per website. However, collecting large-scale and fresh traces is quite cost-consuming and unrealistic. Moreover, these deep-learning-based WF attacks lack flexibility because they require a long bootstrap time for retraining when facing new traffic traces with different distributions or newly added monitored websites. This paper proposes a high-accuracy WF attack, Contrastive Fingerprinting (CF), which leverages contrastive learning and data augmentation over a few training traces. Extensive experiments have validated the accuracy and robustness of the CF attack on challenging datasets, which only collect a few training traces from each website and identify the testing traces with different distributions. For example, when each monitored website only has 20 training traces, CF identifies monitored websites with a high accuracy of 90.4% in the closed-world scenario and distinguishes monitored websites with a high True Positive Rate of 91.2% in the open-world scenario. We also show that CF outperforms two existing WF attacks with few-shot traces and has strong practicability.

## KEYWORDS

Website fingerprinting, User privacy, Tor, Contrastive learning, Few-shot learning

## 1 INTRODUCTION

Anonymity systems like Tor [10] protect sensitive network data and keep Web access private, which routes traffic through relays and hides the ultimate destination. However, traffic analysis technologies such as website fingerprinting (WF) [31] make this user's privacy easily broken. WF attacks can identify the visited websites by analyzing the patterns of encrypted traffic traces. For example, a local adversary passively observes the connection between a user and a Tor entry node and extracts traffic features by exploiting the leaked information like the packet size, the transmission direction, and the timing of requested resources.

From the machine learning perspective, WF is a classification problem of websites, where each website is often regarded as one class. WF attackers train a classifier with traffic traces, extracting the unique traffic fingerprint of websites. The trained classifier determines which class the user's trace belongs to and then identifies which website the user visits. Most researchers evaluate WF's performance in two scenarios: *closed-world* and *open-world*. The closed-world scenario requires users only to access a set of monitored websites while the attacker monitors and trains these websites in advance. However, users may access other regular websites, and Internet websites are too enormous to monitor. Therefore, the open-world scenario is more realistic, where the attacker still monitors

a set of monitored websites, but users can access more websites, including *monitored websites* and *unmonitored websites*.

Primitive WF attacks [5, 13, 16, 36] manually extract traffic features and become vulnerable to WF defenses [3, 4, 12, 22] because these defenses often blur traffic features by injecting dummy packets, fixing packet size, or delaying packet transmission. The state-of-the-art WF attacks, like Deep Fingerprinting (DF) [29] and Var-CNN [2], adopt deep learning to achieve up to 96% accuracy in website identification. Their successes are owed to deep learning classifiers' ability to automatically extract traffic features from large amounts of traces. However, these WF attacks have to spend much time collecting and updating training traces to maintain high accuracy. The prior WF studies mostly ignore the impacts of bootstrap time and different distributions of website traces.

*Bootstrap time.* Bootstrap time is the total time to develop an available classifier, including collecting traces and training the classifier [30]. Previous studies ignored that the time gap between collecting training traces and testing users' traces would affect WF's performance. For example, Juarez et al. found that the accuracy of a WF attack dropped from 80% to 30% using the classifier trained ten days before. One website's 'traffic fingerprint' is time-sensitive because of the dynamic network conditions and occasional updates of webpage content. Some WF attacks frequently collect amass traffic and train an efficient classifier to keep pace with the changes in testing traffic, which increases the bootstrap time and the exposure risk.

*Different distributions of traces.* Previous studies assumed that WF attackers wholly duplicated user's settings, i.e., the network condition and Tor-browser-bundle (TBB) version. WF attacks deploy training and testing in the datasets with the same trace distribution. However, different users may adopt different TBB versions under different network conditions, which causes changes in trace distributions. When the WF attack was trained and tested with the traces collected from different TBB versions, its accuracy dropped from 79% to 12% [17]. Therefore, practical WF attacks shall adapt to the difference in trace distributions.

To be applicabe in real environments, WF attacks need to reduce the number of traces required for training, that is, to conduct website fingerprinting with few training samples. However, the existing WF attacks based on deep learning can not work well with few training samples. As shown in Figure 1, the accuracies of DF and Var-CNN sharply decline to less than 50% when given only ten training traces per website. Likewise, if facing WTFPAD defense, the accuracy will drop below 20%. Recently, researchers studied WF attacks in a few-shot environment requiring only a few training traces per website, such as Triplet Fingerprinting (TF) [30] and Adaptive Fingerprinting (AF) [35]. Both adopt transfer learning to develop a feature extractor for the following few-shot learning. However, TF leverages the Semi-Hard-Negative mining strategy and costs high in training. AF uses an adversarial domain network [33] and fails to overcome the unbalanced number of traces with different distributions. Especially facing the testing traces from

different distributions, AF must retrain the feature extractor, which significantly prolongs the bootstrap time.

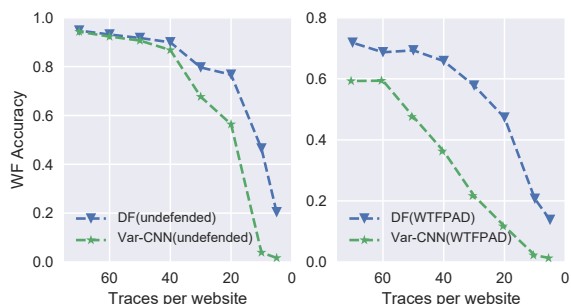

**Figure 1: Accuracy v.s. Number of training traces per website**

In order to overcome the above unrealistic assumptions and weaknesses of WF attacks, we propose *Contrastive Fingerprinting (CF)*, a novel method available in the few-shot scenario. CF utilizes contrastive learning good at feature representation [7, 19] and performs an efficient WF attack through two stages. The pre-training stage obtains a feature extractor, and the few-shot one trains a classifier for testing the user's traces. The main contributions include:

- CF solves the WF's bootstrap time dilemma by decreasing the training traces collected from each website, making CF more practical in real networks. The well-designed data augmentation method can create adequate training traces based on a few collected traces (i.e., the few-show scenario).
- Using contrastive learning, CF applies contrastive loss to build an efficient feature extractor that considers the global features of encrypted traffic. It outperforms two popular few-shot WF methods, Triplet Fingerprinting (TF) and Adaptive Fingerprinting (AF). For example, when collecting and training 20 traffic traces for each monitored website, CF can correctly identify about 85% of testing traces in the closed-world scenario. While with the same training traces, other WFs' accuracy is about 70%. In the open-world scenario, CF also works well in judging whether the user visits a monitored website, and its True Positive Rate is over 91.2%, with the False Positive Rate less than 7.4%.
- We have used four datasets to verify the accuracy and robustness of CF, which were collected at different periods with different TBB versions. Among them, the CF Dataset was collected in 2023 using Tor Browser V12.X. As far as we know, this paper is the first one that considers the timeliness of traces, the influences of browsers, different network conditions, and the impacts of WF defenses under the few-shot scenario. The extensive experiments in the closed-world and open-world scenarios show that CF demonstrates satisfactory performance in identifying websites and strong robustness in dealing with different distributions of traces. For example, CF may achieve 80.1% accuracy against WTF-PAD defense in the closed-world scenario.

## 2 BACKGROUND

### 2.1 Threat Model

This paper considers the same threat model as previous WF studies [5, 13, 16, 36]. As shown in Figure 2, website access goes through one anonymous system like Tor, which allows users to browse the Internet anonymously. An attacker does not manipulate packet transmissions but passively intercepts users' encrypted traces. It may be an Internet service provider (ISP) or a device located in the network path between ISP and the first Tor relay. After training a website classifier by analyzing the unique fingerprint of traces, the attacker can predict the websites visited by users.

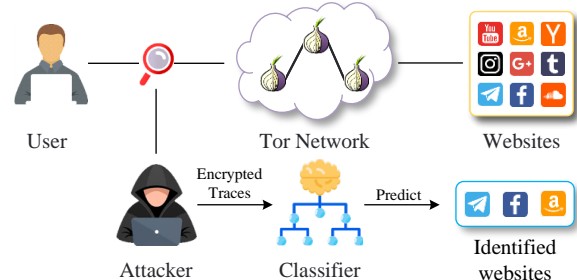

**Figure 2: WF threat model**

### 2.2 Transfer-learning-based WF

Supervised learning methods [6, 38] have achieved good performance based on a large amount of labeled data for training, which is tedious and expensive. However, a traditional supervised method often fails when dealing with the new tested data with different distributions. On the other hand, transfer learning methods [32, 41] recognize the knowledge learned from previous tasks and apply it to novel tasks. Specifically, given a pre-training dataset with lots of labeled training data, transfer learning learns it and applies the transferred knowledge to a new dataset.

Therefore, recent WF attacks [30, 35] (refer to Subsection 2.2.1 and 2.2.2) use transfer learning to identify websites in the few-shot scenarios, which only collect a few traces of each website for training. These few-shot WF attacks usually include three stages: *pre-training*, *few-shot training*, and *testing*. The first stage operates in a *pre-training dataset* with amounts of labeled traces in advance and obtains a feature extractor. The last two operate in a *few-shot dataset* that includes only a few labeled traces of each monitored website (for few-shot training) and amounts of unlabeled traces (for testing), and this dataset is divided into a *support set* and a *query set*. Then, in the few-shot training stage, the trained feature extractor transfers the learned knowledge to train a website-identification classifier using the traces in the support set. Finally, in the testing stage, the trained classifier identifies the traces in the query set and predicts their visited websites.

As one popular transfer learning method, fine-tuning is successful in many fields [40]. First, fine-tuning trains a complete neural network with the pre-training dataset, including many labeled data. Then it achieves the fine-tuned neural network by tuning the last network layer with the new few-shot dataset while freezing most network layers and hyperparameters. However, fine-tuning

requires that the pre-training and few-shot datasets have similar data distribution and scale. This requirement hampers the application of fine-tuning in actual WF attacks, which only collect and label few-shot traces.

*2.2.1 Triplet Fingerprinting.* Triplet Fingerprinting (TF) [30] applies the triplet network [25], which contains three parallel and identical sub-networks with the same weights and hyperparameters. In the pre-training stage, the input of these three sub-networks corresponds to a triplet: Anchor (A), the main reference; Positive (P), another example from A's class; Negative (D), an example selected from any class that except A's class. The three sub-networks take the triplet as inputs, respectively, and update through triplet loss which tries to minimize the distance between traces from the same class (A, P) and maximize the distance for traces from different classes (A, D). After pre-training, TF chooses one sub-network as the feature extractor for the following stages. During the few-shot training, a few collected training traces from the support set are fed into the pre-trained feature extractor to generate the corresponding embedded vector for each website. Finally, these embedded vectors are used to train a k-NN classifier to complete the website classification task in the testing phase.

However, in TF, an anchor sample only considers a single positive and negative sample at a time, which significantly reduces the convergence speed and effectiveness of the model. Therefore, to ensure the effectiveness of feature extraction, TF leverages the Semi-Hard-Negative mining strategy instead of randomly selecting samples. This strategy aims to choose a negative sample D closer to the anchor sample A, which helps the feature extractor identify more valuable features. However, it is time-consuming and increases calculation complexity.

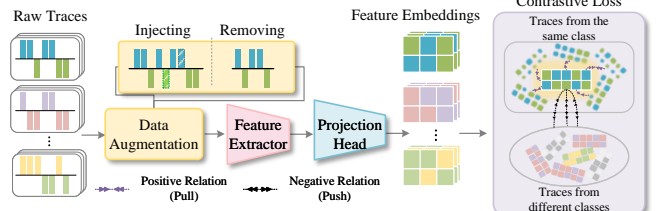

**Figure 3: Contrastive network during the pre-training stage.**

*2.2.2 Adaptive Fingerprinting.* Adaptive Fingerprinting (AF) [35] adopts Adversarial Domain Adaption to extract domain-invariant features. In the pre-training stage, AF consists of a Feature Extractor, a Domain Discriminator, and a Source Classifier. The Feature Extractor takes the pre-training dataset and the query set in the few-shot dataset as inputs and outputs the domain-invariant features. On the contrary, the Domain Discriminator aims to distinguish the right domain of the features generated by the Feature Extractor. The purpose of the Source Classifier is to predict the class of the pre-training dataset with the Feature Extractor's output. In other words, the Source Classifier and Domain Discriminator help to build a more effective Feature Extractor. Like TF, AF freezes the pre-trained Feature Extractor and combines it with a k-NN classifier in the few-shot training and testing stages.

Besides the pre-training dataset, AF also introduces the few-shot dataset to generate domain-invariant features in the pre-training stage. However, although the few-shot traces used in the pre-training stage are unlabeled, this practice is still unrealistic. Specifically, when facing a new dataset with another distribution, the feature extractor needs to be re-trained, significantly extending bootstrap time and making AF unpractical. Moreover, using a traditional domain adversarial network, AF fails to overcome the unbalanced sample size among datasets from different distributions. Therefore, when the number of training traces per website declines to less than 10, AF's accuracy of website identification drops sharply.

In order to solve the above problems of TF and AF, this paper combines contrastive learning and data augmentation technologies to design an efficient few-shot WF attack Contrastive Fingerprinting (CF). Contrastive learning is an advanced feature extraction method that substantially improves the quality of the learned representations [7], while data augmentation efficiently provides the supplement traces for training. Section 4 will evaluate CF's performance and show its superiority compared with TF and AF.

## 3 METHOD

WF attacks generally represent a traffic trace as a sequence of tuples. Each tuple $< \pm packet\_size >$ records the information of one packet whose length is $packet\_size$. The sign of $packet\_size$ indicates the direction: positive (+) represents outgoing/transmission, and negative (−) represents incoming/reception. Previous studies [2, 29] show that the value of $packet\_size$ does not contribute to accuracy, so we set it as a unified value $packet\_size = 1$.

Next, we design *Contrastive Fingerprinting (CF)* and present the detailed structure of the contrastive network.

### 3.1 Workflow of CF Attack

As a transfer learning method, the design of CF attack consists of three stages: *pre-training*, *few-shot training*, and *testing*. The details of each stage are described as follows.

**Pre-training stage.** As shown in Figure3, during the pre-training stage, we train a contrastive network with pre-training dataset via four components: data augmentation, feature extractor, projection head, and contrastive loss. We use data augmentation to enrich traces efficiently. The feature extractor and projection head are two neural networks updated through contrastive loss. Thus, the goal of the pre-training stage is to obtain a feature extractor that can quickly adapt to any dataset to complete subsequent few-shot classification tasks. We will discuss each part of the contrastive network for details in Section3.2.

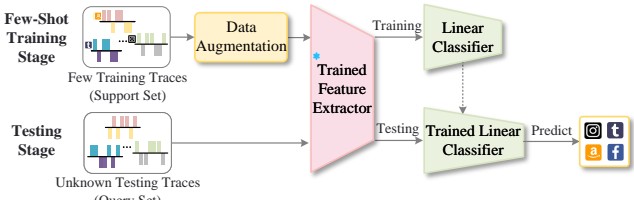

**Figure 4: The few-shot training and testing stages of CF.**

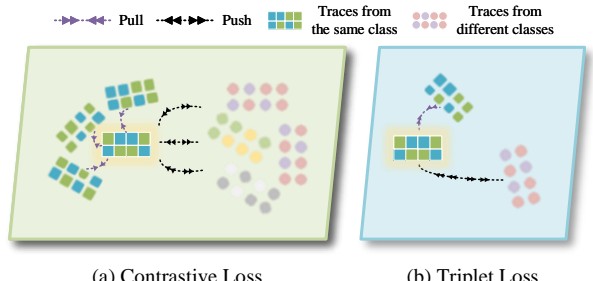

(a) Contrastive Loss          (b) Triplet Loss

**Figure 5: The calculation of contrastive loss and triplet loss.**

**Few-shot training stage.** In the few-shot training stage, the attacker leverages the support set of the few-shot dataset (e.g., five traces for each website in 5-shot training), which have a different distribution from the pre-training dataset. As shown in Figure4, at first, data augmentation is deployed on the few training traces from the support set to increase the diversity of samples. After that, these traces are fed into the trained feature extractor to generate corresponding feature embeddings for each website. It is worth noting that all the structure and parameters of the trained feature extractor always stay the same as the model obtained in the pre-training phase (use * to mark in Figure4). The feature embeddings are then used to train a linear classifier for the later testing based on the query set. After several experiments, we find that the linear classifier performs better than other classifiers, like k-NN, SVM, MLP, etc.

It is worth noting that we do not adopt the projection head in the following few-shot training and testing stages. Based on previous work related to contrastive learning[7, 19], the projection head helps the hidden layer before it to learn a better representation. Instead, the representation quality of the layer after the projection head is weakened. We will also prove that for WF attacks in Section4.2.

**Testing stage.** In the testing stage, the same as the few-shot training stage, the unknown testing traces from the query set (without label) captured by the attacker are fed into the trained feature extractor to get several feature embeddings. After that, the linear classifier trained (during the few-shot training stage) predicts the label of feature embeddings of the unknown traces.

## 3.2 Contrastive Network

During the pre-training phase, the contrastive network learns feature representations via two neural networks updated by contrastive loss, which contains four major components. Given an input batch of traces, contrastive networks first apply two *data augmentation* methods to obtain two copies of the batch. Next, both copies are propagated forward through a *feature extractor* and a small neural network *projection head* to obtain a 128-dimensional feature embedding. Finally, the *contrastive loss* is computed on the outputs of the *projection head*. The complete structure of the contrastive network is shown in Figure3.

*3.2.1 Data Augmentation.* Data augmentation[27] is widely used in various fields[21] and shown to generalize and enhance classification performance in few-shot scenarios. It has been verified that adding the training images of reality disturbances such as occlusion

and rotating helps build a more robust model in computer vision[39]. Thus, we design two data augmentation manipulations: *Injecting* and *Removing* by adding the possible real disturbances in the network into traffic. *Injecting* adds background traffic into a collected raw trace $t$ by randomly injecting +1 or −1 into a random position. On the contrary, *Removing* chooses a position randomly and deletes its corresponding packets. In particular, the *Injecting* manipulation introduces background traffic when accessing other applications or browser tabs. *Removing* simulates the packet loss and packet retransmissions due to network congestion or transmission errors. The introduction of disturbances enhances the network's ability to infer the traffic context reasonably, making the network more robust to dynamic network conditions, thereby mitigating the impact of excessive bootstrap time on WF attacks.

For each input trace, $t$, we introduce a certain level of random disturbance into $t$ by generating the above two augmentations, $\widetilde{t} = Aug(t)$, while the augmented trace keeps the website label unchanged. Obviously, $\widetilde{t}$ includes two traces, one is after injecting and the other is after removing manipulation. Therefore, for a set of $N$ randomly sampled trace/label pairs, $\{t_k, y_k\}_{k=1\ldots N}$, the corresponding batch used for the following training consists of $2N$ pairs, $\{\widetilde{t_l}, \widetilde{y_l}\}_{l=1\ldots 2N}$ where $\widetilde{t}_{2k}$ and $\widetilde{t}_{2k-1}$ are two data augmentations of $t_k$ ($k = 1\ldots N$) and $\widetilde{y}_{2k} = \widetilde{y}_{2k-1} = y_k$.

*3.2.2 Feature Extractor and Projection Head.* These two are both neural networks. Feature Extractor $Enc(\cdot)$, a deep neural network, is based on CNN with convolutional layers, batch normalization layers, max pooling layers, and dropout layers. We will describe the model chosen and the reasons for it in the Section 3.3. The feature extractor extracts representation vectors, $r = Enc(\widetilde{t})$ from augmented traces. Both augmented samples are separately input to the feature extractor, resulting in a pair of representation vectors.

After that, a small neural network, projection head $proj(\cdot)$ maps representations to the space where contrastive loss is applied. The outputs of the projection head, $e = proj(r)$, are normalized to the unit hypersphere with the size of 128, which enables using an inner product to measure distances in the projection space. According to previous studies[7, 19], adding a nonlinear projection head introduces a positive impact compared to computing contrastive loss directly on $r$, which will also be proved for WF attacks in Section 4.2. Obviously, the $2N$ pairs after data augmentations, $\{\widetilde{t_l}, \widetilde{y_l}\}_{l=1\ldots 2N}$ corresponds to $2N$ embedding/label pairs $\{\widetilde{e_l}, \widetilde{y_l}\}_{l=1\ldots 2N}$ in the projection space.

*3.2.3 Contrastive Loss.* The contrastive loss is computed on the outputs of the projection head. Randomly select a feature embedding from the $2N$ embedding/label pairs, $e_k$, in the projection space, the contrastive loss tries to narrow the distance between $e_k$ and the embeddings (traces) with the same label from the whole training batch. Conversely, the distance between $e_k$ and the traces from different classes in this $2N$ training batch will be extended. In other words, after computing contrastive loss, the traces from the same class will have positive relations, which will pull them together. Also, the negative relations will push away the traces from different websites in the projection space.

Figure 5 (a) illustrates the calculation process of contrastive loss in which all the traces in the whole training batch are included. The distance between the selected and the same class traces is

**Table 1: CF's Hyperparameter tuning, the pre-training stage.**

| Hyperparameters | Search Space | Final Value |
|---|---|---|
| Feature Extractor's Model | GoogleNet, ResNet, Var-CNN, DF | DF |
| Projection Head's Model | MLP, Linear | MLP |
| Percent of Data Augmentation | [0 ... 1] | 0.1 |
| Feature Embedding ($r$)'s Size | [64 ... 512] | 256 |
| Feature Embedding ($e$)'s Size | [64 ... 512] | 128 |
| Batch Size | [32 ... 256] | 64, 128 |
| Similarity Metrics | Euclidean, Cosine | Cosine |

shortened. Simultaneously, the traces from different classes are pushed away from the selected trace. However, as discussed in Section 2, triplet loss used by TF [30] uses only one *positive* (from the same class) and one negative (from different classes) trace for the selected trace, as shown in Figure 5 (b). Global consideration of the entire training batch of traces allows our model with contrastive loss to achieve state-of-the-art performance. Without the need for *Semi-Hard-Negative* mining used by triplet loss, CF reduces the cost of pre-training and further increases the feasibility.

## 3.3 Hyperparameter Tuning

To evaluate and select the hyperparameters for CF, we extensively search through the hyperparameter space. As shown in Table 1, the primary hyperparameters that we tune, the candidates' range, and the final value we select are listed.

**Base Model.** There are two base models for developing CF during the pre-training stage, as shown in Figure 3. For the feature extractor, we test with various standard neural network models for computer vision, including ResNet, GoogleNet, and two customized WF models, DF [29] and Var-CNN [2]. Experiments find that the feature extractor with the DF model achieves a better balance between classification accuracy and training cost than other candidates. According to the previous studies based on contrastive loss, we select an MLP with one hidden layer as the base model for projection head instead of a linear projection, which we will explain in detail in Section 4.2.

**Percent of Data Augmentation.** The percent of data augmentation represents the ratio of manipulated data to the length of trace $t$. We find that the propitiate percent for *Injection* and *Removing* are both located in the range of [0.05, 0.15]. When the percentage grows above 0.15, the performance of models gradually decreases. Since the percent of data augmentation is restricted to a smaller range, the models can understand the content of the trace in a global sense rather than a local sense which may not always exist in all traces. Furthermore, training with these augmented traces makes the framework more robust to dynamic network conditions and traces from different distributions.

**Feature Embedding's Size.** The feature embeddings' size refers to the last dense layer of feature extractor ($r$) and projection head ($e$). After extensive experiments, we find that the changes in size do not have a significant impact on the final accuracy. Thus, we follow the original output size of the DF model $r = 256$. Furthermore, for $e$, which is used for contrastive computing loss, we select 128 as the final size to speed up the calculation.

**Others.** We choose other hyperparameters based on our preliminary results. We find that Cosine distance is more advantageous in distinguishing burst patterns in traffic than Euclidean distance, with 0.9–2.0% better accuracy. Also, batch size = 64 and batch size = 128 can provide 1.5% better accuracy than other candidates.

## 4 EXPERIMENTAL EVALUATIONS

In this section, we design several demanding and challenging evaluations to show the efficiency of CF under few-shot scenarios, which can shorten the bootstrap time and reduce the adverse impact of different-distribution traces on the performance of WF attacks. Also, we compare our method with previous few-shot WF attacks.

### 4.1 Experiment Setup

*4.1.1 Dataset.* We adopt four datasets (Table 8) collected at different periods with different TBB versions to investigate the accuracy and robustness of CF. Most importantly, these comprehensive datasets consider the timeliness of traces, the influences of browsers, and network conditions under the few-shot scenario.

The first is **CF Dataset (ours)**, whose traces were collected between October 2022 and February 2023 using Tor Browser V12.X. We used *tor-browser-crawler* to drive the Tor Browser, allowing more realistic crawls than *wget* and *curl*. The monitored websites come from the top 500 Alexa websites, and the unmonitored websites come from the top 10,000 Alexa websites. The URLs of monitored and unmonitored websites do not coincide. In order to ensure the validity of the corrupted traces, we also check the packet length and access status of the collected traces to eliminate the traces that failed to access the target websites. CF's subsets are as follows:

- CF500: 500 monitored websites, each has 100 traces.
- CF9000: 9000 unmonitored websites, each has 1 trace.

Other 3 datasets are the Wang[36], AWF[24] and DF[29] datasets.

**Wang Dataset** was collected in 2013 using TBB V3.X. The *monitored websites* come from a list of sites blocked in China, the UK, and Saudi Arabia. The *unmonitored websites* come from Alexa's top websites. We divide it into two subsets:

- Wang100: 100 monitored websites, each has 90 traces.
- Wang9000: 9000 unmonitored websites, each has 1 trace.

**AWF Dataset** was collected in 2016 using TBB V6.5. All websites come from Alexa's top websites, but their URLs do not intersect. There are three subsets as follows:

- AWF100: 100 monitored websites, each has 90 traces.
- AWF775: 775 monitored websites, each has 25 traces.
- AWF9000: 9000 unmonitored websites, each has 1 trace.

**DF Dataset** was collected in 2016 using TBB V6.X, whose websites also come from Alexa's top websites. Two subsets are as below:

- DF95: 95 monitored websites, each has 100 traces.
- DF9000: 9000 unmonitored websites, each has one trace.

*4.1.2 Metrics.* We follow the metrics used in the paper [2]. In the closed-world scenario, WF attacks address a multi-classification problem that measures performance by Accuracy, the ratio of the number of monitored websites correctly identified to the number of testing traces. In the open-world scenario, WF attacks distinguish whether the testing traffic is from a monitored or an unmonitored

website and use two metrics: Positive Rate (TPR) and False Positive Rate (FPR). TPR means the proportion of the monitored website traces correctly classified as the monitored class. FPR refers to the proportion of the unmonitored website traces misclassified as the monitored class.

## 4.2 Closed-World Evaluation

In the closed-world scenario, we adopt different datasets for the pretraining and few-shot training stages. With a small amount of training traces, we inspect the ability of WF attacks to adapt to different distribution datasets. In subsequent closed-world experiments, CF500 and AWF775, collecting traces from more than 500 monitored websites, will be used in the pretraining stage to train the feature extractor. Other datasets, including fewer monitored websites, will be used in the few-shot stage. The reason is that contrastive learning is to complete subsequent classification tasks by comparing the differences between classes and measuring the distance between classes. In pre-training the feature extractor, more classes can help the model obtain a more robust ability for feature extraction.

Moreover, the dataset used in the few-shot stage is divided into a support set (for few-shot training) and a query set (for testing). We use $N = \{1, 5, 10, 15, 20\}$ to denote the number of training traces per website in the support set, and these traces are used in the few-shot stage for training. The number of testing traces per website in the query set is 20. To more comprehensively examine the accuracy of WF attacks, we ran each experiment ten times and recorded the mean and standard deviation.

**Experiment 1: Impacts of $N$, the number of training traces per website in the few-shot training stage.**

Table 2 records three dataset combinations: CF500-Wang100, CF500-AWF100, and CF500-DF95, where CF500 is used in the pre-training stage (called the pre-training dataset), and Wang100, AWF100, and DF95 are used in the few-shot stage (called the few-shot dataset), respectively. This experiment compares CF, TF, AF, and the Fine-tuning method in the close-world scenario when the number of training traces per website in the few-shot training stage changes by $N = 5, 10, 15, 20$. Most importantly, we study the impacts of $N$ on WF's accuracy, whose results are recorded in the form of mean ± standard deviation.

We observe that as $N$ grows, four WF methods perform better. Our CF consistently achieves the highest accuracy with $N \leq 20$ training samples when facing different dataset compositions. For example, with $N = 10$ training traces per website, the accuracy of the Fine-tuning method is less than 75.4%, whereas CF's accuracy is about 85%. Surprisingly, CF's accuracy with $N = 20$ (the 20-shot learning) on the CF500 - Wang100 dataset gradually increases to 90.4%. Even in the cases with $N = 5$, CF is the only method whose accuracy is over 80% on all datasets whose traffic has different distributions. These results show that CF is suitable for the few-shot scenario.

Furthermore, the superiority of our CF is more significant under challenging conditions, i.e., using fewer training traces. Figure 6 presents the accuracy gap, which records the accuracy difference between CF and another method. The higher the accuracy gap, the more superiority of CF. The left subfigure shows that the accuracy

gap curves of the Fine-tuning method increase when $N$ decreases on three datasets with different distributions. For example, on the CF500-AWF100 dataset, the accuracy gap between CF and Fine-tuning is nearly 10% with $N = 10$. In comparison, it is nearly 20% with $N = 5$. AF's curves of the accuracy gap in the right subfigure have similar trends. However, AF's accuracy drops faster when $N = 1, 5$, especially on the CF500-DF95 dataset. AF's authors have explained the reason: AF lacks sufficient traces to pre-train a good Domain Discriminator when N is extremely small.

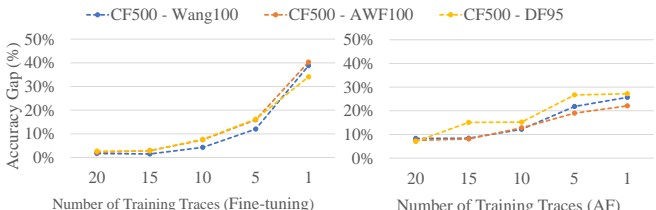

**Figure 6: The accuracy gap (%) between CF and other methods.**

**Experiment 2: Impacts of $M$, the number of training traces per website in the pre-training stage.**

We study whether the number of training traces per website in the pre-training stage affects WF attacks. Here, we select AWF100 and DF95 as the pre-training and few-shot datasets, respectively. We use $M = \{25, 50, 100, 150, 200\}$ and $N = \{10, 15\}$ to represent the number of training traces per website in the pre-training and few-shot stages. Table 3 illustrates that the accuracy of all WF methods increases when $M$ increases. The accuracy of CF and AF gradually stabilized after $M > 50$. The TF and fine-tuning methods require a large value of $M$ (e.g., $M = 100$) during the pre-training stage, significantly extending the bootstrap time. Among all permutations of $M$ and $N$, CF exhibits the most superior accuracy, consistent with the results in Table 2.

**Experiment 3: The role of data augmentation.** We are the first to combine contrastive learning and data augmentation for WF attacks. As shown in Table 6, data augmentation enhances CF's attack accuracy. Surprisingly, the data augmentation method improves the accuracy more when the number of training traces per website is smaller. This observation indicates that our data augmentation method is practical to improve the diversity of training samples in few-shot learning. Combining Table 2 and Table 6, we note that even without data augmentation, CF's accuracy is better than the other three methods, which also illustrates the superiority of contrastive network in few-shot WF attacks.

**Experiment 4: The role of a nonlinear projection head.**

We then study the role of the projection head, $e = proj(r)$. In Section 3.3, we choose an MLP with one hidden layer as the base model for the projection head. We evaluate three structures for this projection head: identity mapping (no projection head), linear projection, and nonlinear projection with one additional hidden layer (and ReLU activation). We observe that a nonlinear projection is better than a linear projection (+5%) and much better than no projection head (>15%). This finding is consistent with [7]. Furthermore, the layer before the projection head performs much better when using the nonlinear projection, as shown in Table 6.

Table 2: Closed-World: The impacts of $N$ on WF accuracy (%) on the datasets with different distributions

| Pre-training | Few-shot Training & Testing | Method | $N = 1$ | $N = 5$ | $N = 10$ | $N = 15$ | $N = 20$ |
|---|---|---|---|---|---|---|---|
| CF500 | Wang100 | Fine-tuning | 43.4±2.2 | 62.8±1.7 | 75.4±1.3 | 80.4±1.0 | 82.0±1.1 |
| | | TF | 65.9±1.9 | 79.0±0.5 | 80.7±1.0 | 81.6±0.6 | 81.9±0.6 |
| | | AF | 30.2±4.2 | 72.3±2.2 | 83.3±1.7 | 87.2±1.3 | 88.7±0.8 |
| | | **CF (ours)** | **64.4±0.2** | **84.3±0.1** | **87.6±0.2** | **88.7±0.1** | **90.4±0.1** |
| CF500 | AWF100 | Fine-tuning | 42.3±1.9 | 61.0±1.0 | 72.7±0.9 | 79.2±0.8 | 80.9±0.4 |
| | | TF | 60.9±1.8 | 72.1±0.9 | 79.4±0.6 | 80.2±0.6 | 81.5±0.4 |
| | | AF | 24.1±2.9 | 64.1±2.7 | 78.0±1.1 | 84.4±1.1 | 86.2±0.9 |
| | | **CF (ours)** | **64.4±0.2** | **80.0±0.1** | **85.5±0.1** | **87.3±0.1** | **88.4±0.1** |
| CF500 | DF95 | Fine-tuning | 28.2±2.0 | 51.6±2.9 | 68.5±1.4 | 69.8±2.0 | 80.1±0.1 |
| | | TF | 45.9±3.4 | 64.5±1.6 | 68.0±0.9 | 69.8±1.2 | 70.3±1.7 |
| | | AF | 21.3±2.8 | 62.1±2.0 | 76.0±1.3 | 82.1±1.2 | 83.4±1.0 |
| | | **CF (ours)** | **55.4±0.2** | **78.3±0.1** | **83.7±0.1** | **84.9±0.1** | **87.1±0.1** |

Table 3: Closed-World: Impacts of $M$ and $N$ on WF accuracy (%).

| Pre-training | Few-shot Training & Testing | Method | $M = 25$ | | $M = 50$ | | $M = 100$ | | $M = 200$ | |
|---|---|---|---|---|---|---|---|---|---|---|
| | | | $N = 10$ | $N = 15$ | $N = 10$ | $N = 15$ | $N = 10$ | $N = 15$ | $N = 10$ | $N = 15$ |
| AWF100 | DF95 | Fine-tuning | 48.9±5.3 | 55.7±2.5 | 59.7±4.7 | 62.5±1.3 | 68.3±1.2 | 70.6±2.2 | 68.1±1.3 | 70.2±1.9 |
| | | TF | 60.8±1.1 | 62.3±1.0 | 61.7±1.6 | 62.9±1.1 | 69.4±0.9 | 71.0±0.5 | 69.9±1.3 | 71.1±0.6 |
| | | AF | 76.1±1.5 | 79.9±1.1 | 75.4±1.7 | 80.7±2.0 | 76.6±1.8 | 81.8±1.6 | 75.6±1.5 | 81.5±1.4 |
| | | **CF (ours)** | **79.6±0.1** | **82.7±0.2** | **81.5±0.1** | **84.6±0.1** | **81.4±0.1** | **84.0±0.1** | **81.9±0.1** | **84.2±0.1** |

Table 4: Closed-World: WF accuracy (%) on defended datasets.

| Method | $N = 5$ | $N = 10$ | $N = 15$ | $N = 20$ |
|---|---|---|---|---|
| Fine-tuning | 37.1±1.2 | 58.2±1.4 | 61.2±1.2 | 61.9±1.0 |
| TF | 50.5±1.0 | 58.3±1.2 | 60.9±1.0 | 61.3±0.7 |
| AF | 38.9±3.9 | 60.9±2.3 | 66.9±1.6 | 76.9±1.3 |
| **CF (ours)** | **60.1±0.2** | **68.4±0.2** | **77.8±0.2** | **80.1±0.3** |

This experiment uses CF500 and Wang100 as the pre-training and few-shot training datasets, respectively. We observe that the model without the projection head performs much better than that with the projection head (>10%). $e = proj(r)$ may remove helpful information for classification. More information can be formed and maintained by leveraging the nonlinear transformation $proj(\cdot)$. Therefore, CF only applies the feature extractor instead of combining the feature extractor and projection head in the few-show training and testing stages.

**Experiment 5: Evaluations under WTFPAD defense.** We further evaluate CF under WTFPAD defense [18], which is a leading candidate to be applied in Tor due to its low overhead. However, generating defended traces with WTFPAD needs timestamps of packets unavailable in the AWF100 dataset (with packet direction only). Then, We only reproduce the WTFPAD defense on CF500 and Wang100 to obtain their corresponding defended datasets with the same scale. The number of traces per website $N$ for training and $T$ for testing is consistent with Section 4.2.

Table 4 compares CF and other methods under the WTFPAD defense. We choose CF500 as the pre-training dataset and Wang100 as the few-shot dataset while evaluating each method with $N = \{1, 5, 10, 15\}$ training traces per website. The results show that the

attacks' accuracy significantly decreases compared to the non-defended dataset in Table 2. However, CF can reach 60% accuracy with only = 5 training traces and outperforms other methods. Moreover, CF is the only attack with over 80% accuracy against the WTFPAD defense.

The above experiments in the closed-world scenario fully demonstrate CF's superiority in accuracy. It is worth noting that the fluctuation of the accuracy of each method is inevitable in a few-shot scene. For example, the standard deviations of accuracy are above 1.0% in the fine-tuning, TF, and AF attacks. Moreover, the accuracy fluctuates even more violently when there is less training data or facing defended datasets. Even so, in the face of different datasets and few training traces, CF has always remained stable accuracy with a low standard deviation of accuracy (less than 0.5%). As mentioned above, instead of using the traditional k-NN classifier, we choose the linear classifier for few-shot training and testing. Compared to k-NN, which calculates similarity based on distance, the linear classifier tends to be more stable after few-shot training, resulting in more minor accuracy fluctuations.

## 4.3 Open-World Evaluation

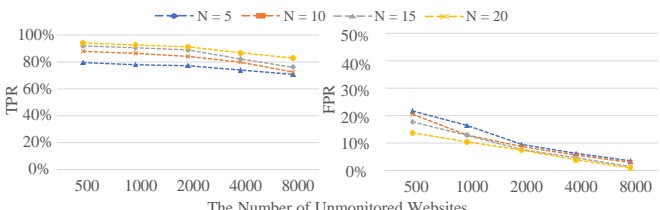

Figure 7: TPR and FPR v.s. Number of unmonitored websites.

**Table 5: Open-World: The impact of number of training traces per website on TPR and FPR (%).**

| Pre-training | Few-shot Training & Testing | Method | N = 1 | | N = 5 | | N = 10 | | N = 15 | | N = 20 | |
|---|---|---|---|---|---|---|---|---|---|---|---|---|
| | | | TPR | FPR | TPR | FPR | TPR | FPR | TPR | FPR | TPR | FPR |
| {CF500, CF9000} | {DF95, DF9000} | Fine-tuning | 2.2 | 1.9 | 66.1 | 12.3 | 79.9 | 9.2 | 83.3 | 8.1 | 85.6 | 8.2 |
| | | TF | 34.2 | 24.2 | 71.7 | 22.0 | 74.6 | 15.9 | 76.7 | 14.9 | 77.3 | 10.6 |
| | | **CF (ours)** | **48.9** | **10.1** | **77.2** | **9.5** | **84.1** | **8.7** | **88.9** | **7.6** | **91.2** | **7.4** |

**Table 6: Impacts of projection head and data augmentation**

| Model during Few-shot Training & Testing Stage | N = 5 | N = 10 | N = 15 | N = 20 |
|---|---|---|---|---|
| With Projection Head | 66.3±0.1 | 74.3±0.1 | 78.9±0.2 | 80.8±0.1 |
| Without Data Augmentation | 80.5±0.2 | 84.8±0.2 | 85.8±0.1 | 87.9±0.1 |
| **Selected Model** | **84.3±0.1** | **87.6±0.2** | **88.7±0.1** | **90.4±0.1** |

In the open-world scenario, the attacker shall determine whether the user is visiting a monitored or unmonitored website. We use $N = \{1, 5, 10, 15, 20\}$ and $T = 20$ to denote the number of traces per monitored website in the support set (for few-shot training) and the query set (for testing), respectively. Like the previous works [30, 35], considering a balance between two classes, we set the number of traces for the unmonitored website as $N' = \{100, 500, 1000, 1500, 2000\}$ and $T' = 2000$. The metrics for open-world evaluations are TPR and FPR. Here, we use {CF500, CF9000} to pre-train the feature extractors of each method, where the CF500 dataset is for the monitored websites, and CF9000 is for the unmonitored websites. To test the effectiveness of the methods in the different distribution scenarios, we choose {DF95, DF9000} as the few-shot dataset.

Moreover, as discussed in Table 6, we retain the selected model (with data augmentation and without projection head) for open-world evaluation. Since AF does not provide the code and complete experimental results of the open-world scenario, we only compare CF with TF and the fine-tuning method.

**Experiment 1: Impacts of the number of training traces per website during the few-shot training stage.** As shown in Table 5, the TPR tends to increase with the reduction of FPR for all the WF attacks when the number of training traces during the pre-training stage increases. Somewhat different from the experimental results in the closed-world scenario, the accuracy of the fine-tuning method outperforms the TF attack in most cases in the open-world. The TF attack may fail because adding unmonitored websites makes it difficult for the TF feature extractor iterated by the triplet loss to extract helpful information from the more complex class composition.

At the same time, CF with contrastive loss overcomes the adverse effects of the more challenging open-world scenario and maintains its validity. The results show that the CF attack consistently achieves higher TPR and lower FPR than other methods. With 10-shot learning, CF can reach 84.1% TPR and 8.7% FPR, while other methods can only attain TPR under 80% and over 10% FPR. When the number of training traces reaches $N = 20$, the TPR of the CF attack increases to 91.2%, and the FPR is reduced to only

7.4%. Our method consistently performs best on TPR and FPR in all open-world experimental settings.

**Experiment 2: Impacts of the open-world scale.** It is interesting to explore the performance of CF on a sizeable open-world scale, considering more unmonitored sites. Same as in Section 4.3, we use {CF500, CF9000} as the pre-training dataset and {DF95, DF9000} as the few-shot dataset. CF500 and DF95 are for the monitored websites, and CF9000 and DF9000 are for the unmonitored websites. Then, we evaluate the CF attack against the different numbers of unmonitored sites: $N'$=500, 1K, 2K, 4K, and 8K on the DF9000 dataset. Figure 7 demonstrates that for all values of $N$, the TPR tends to decrease with the reduction of FPR when the open-world scale increases. For example, when dealing with 8K unmonitored sites, CF's TPR drops to less than 80% with $N = 5$ training traces per website. However, against moderate-sized unmonitored sets (e.g., 4K unmonitored websites), CF can still guarantee the effectiveness of the WF attack with over 85% TPR and less than 7% FPR when training traces for each website $N >= 10$.

## 5 CONCLUSION AND FUTURE WORK

We propose Contrastive Fingerprinting (CF), leveraging contrastive learning and data augmentation for few-shot learning in WF attacks. CF allows a WF attacker to achieve high accuracy under closed-world and open-world scenarios using only a few training traces. The evaluation results show that CF can maintain over 90% accuracy even when the pre-training and few-shot datasets are from different distributions (collected a few years apart). Compared with two existing few-shot WF attacks, CF achieves the highest accuracy when $N$ is less than 20, demonstrating the robustness of our well-designed contrastive network. Moreover, even under open-world settings, CF outperforms previous works with 91.2% TPR. These results illustrate that CF significantly improves WF accuracy in the real world using relatively few training traces and low computational costs.

However, like most WF attacks, CF considers that a user only opens one tab to browse websites sequentially and does not deal with the overlaps of traffic traces [17]. However, this single-tab browsing behavior goes against users' habits of opening two or more browsing tabs at once [34]. Some researchers proposed multi-tab WF attacks [8, 9, 37] for classifying websites after splitting traffic, but their jobs depended on a large amount of training traces. Further, how to address both multi-tab browsing and few-shot training challenges would be our future work.

## A DISCUSSION

This section provides a complementary discussion of the effectiveness of our method.

*The same distribution of traces.* Since this paper focuses on the ability of CF and other few-shot WF attacks to quickly migrate to a new data set after a small amount of training data, we mainly explore the classification accuracy of datasets with *different distributions* in the pre-training and testing stages. In order to have a more comprehensive validation of CF, we also conduct the same-distribution evaluation in which the attacker pre-trains the feature extractor on one dataset and performs classification on another same-distribution dataset with different classes. More precisely, we choose AWF775 as the pre-training dataset and AWF100 as the few-shot dataset. These two datasets have a similar distribution in that they were both collected with the same version of TBB (6.X), but the websites' URLs in these two datasets are mutually exclusive. Except for the choice of datasets, other experimental settings are the same as Section 4.2.

Table 7 shows the accuracy of each method under the same-distribution dataset. Compared with the results under datasets from different distributions in Table 2, we notice that the accuracy of each method has been improved, which indicates that excessive bootstrap time or differences in TBB versions will affect the accuracy of website fingerprinting attacks. When facing traces from the same distribution, we find that the gap in performance between the fine-tuning method and several other methods has narrowed. As in the analysis in Section 2.2, to remain effective, fine-tuning requires the pre-training and few-shot datasets to be as close as possible. Additionally, unlike the results in Table 2, the classification accuracy of AF is inferior to that of TF on the same-distribution dataset. We speculate that is because AF uses the *Domain Discriminator* during the pre-training stage. When the number of categories contained in the pre-training dataset and few-shot dataset is imbalanced (e.g., 775 classes in the pre-training dataset while only 100 classes in the few-shot dataset), the *Domain Discriminator* cannot be well trained, which ultimately affect the performance.

As we see in Table 7, CF still outperforms the other two attacks over all cases when facing datasets from the same distribution. With only $N = 5$ training traces per site, CF remains fairly effective with over 90% accuracy. As $N$ grows to 20, the accuracy of CF reaches nearly 95%. Overall, it seems that CF can always be a more reliable option for the WF attacker, no matter in the face of the same distribution dataset or different distribution datasets.

**Table 7: Closed-World: The impacts of $N$ on WF accuracy (%) on the datasets with the same distribution.**

| Method | $N = 1$ | $N = 5$ | $N = 10$ | $N = 15$ | $N = 20$ |
|---|---|---|---|---|---|
| Fine-tuning | 43.3±1.8 | 68.0±0.9 | 79.7±0.9 | 85.0±0.8 | 85.7±0.5 |
| TF | 70.3±2.4 | 86.2±1.0 | 88.1±0.7 | 88.8±0.5 | 89.4±0.4 |
| AF | 27.1±2.7 | 69.1±2.8 | 83.0±0.9 | 87.4±1.1 | 89.2±0.7 |
| **CF (ours)** | **74.4±0.2** | **90.0±0.1** | **92.5±0.1** | **93.3±0.1** | **94.0±0.1** |

## B  RELATED WORK

### B.1  WF Attacks

Prior WF attacks depended on hand-crafted features. Herrmann et al. [15] are the first to examine the WF attack against Tor. By relying on the feature of packet length, he only achieved 3% attack accuracy in the closed-world scenario. After introducing machine learning technologies and carefully selected features, the k-NN attack [36] with k-nearest neighbor (k-NN) classifier, the CUMUL attack with SVM classifier [23], and the Ha-kFP attack with random forest classifier [13] showed better performance against Tor. These WF attacks can achieve over 90% accuracy with 90 training traces per website in the closed-world scenario. Although these hand-crafted methods can work well in some cases, they still have the problem of relying too much on their feature set. Once anonymous systems or defense methods obscure relevant features, the effectiveness of these methods will be minimal.

With the emergence and success of deep learning [20] in various fields, WF attacks with deep learning have gradually been applied [1, 2, 29] for their ability to extract features automatically. In the earlier work, Abe and Goto first applied Stacked Denoising Autoencoders (SDAE) in WF attacks [1]. It can achieve over 88% accuracy with thousands of training traces per website. Recent state-of-art WF attacks are Deep Fingerprinting (DF) [29] and Var-CNN [2]. Inspired by modern image classification networks VGG [28], DF explored 1D convolutional neural networks for network traffic. With a sophisticated and deep CNN architecture, DF achieved superior performance even under the open-world scenario and WTFPAD defenses [18] with thousands of training traces. Var-CNN was designed based on ResNet [14] while importing dilated convolution to replace normal convolution. It can attain 97.8% accuracy in the closed-world scenario with hundreds of training traces per website. However, as discussed in Section 1, when the number of training traces per website decreases to 20 or less, both of these advanced WF attacks fail. The accuracy of these two methods is below 80% and even drops to 50% under the WTFPAD defense.

In order to alleviate the problem of WF attacks relying on large-scale training data, Triplet Fingerprinting (TF) [30] and Adaptive Fingerprinting (AF) [35] have been proposed. Both methods adopt transfer learning to develop a feature extractor for following few-shot learning. However, these two solutions still suffer from several limitations. We have a detailed discussions in Section 2 and compare them with CF to conduct a complete evaluation.

### B.2  WF Defenses

In order to defend against WF attacks, the WF defenses aim to obfuscate unique traffic patterns and decrease the recognition accuracy of websites committed by the adversary. For example, BuFLO [11] proposed by Dyer et al., is the first to restrict traffic transmission to a constant rate with fixed-size packets, thus making traffic features less distinctive. Furthermore, in order to optimize the original design of BuFLO, Tamaraw [4] flexibly determines the amount of padding based on the webpage's total size. CS-BuFLO [3] is sketched as a practical version of BuFLO with new-added congestion sensitivity and rate adaption. Even so, the bandwidth overhead for CS-BuFLO is still over 100%. These constant-rate padding defenses traded excessive bandwidth and latency overheads for moderate security. Specifically, the average page loading time increases two to four times.

Recently, a lightweight countermeasure has been proposed with low latency overhead: WTFPAD [18]. With *Adaptive padding* [26],

                                                          

WTFPAD saves bandwidth by detecting the delays between consecutive bursts and adding dummy packets only when the channel utilization is low. Without delaying any packet, WTFPAD brings moderate bandwidth overhand and no latency overhead, making it a candidate to be applied in Tor [1]. Despite its low cost, the evaluations of WTFPAD show that the defense can significantly reduce the accuracy of WF attacks to below 30%. We evaluate the robustness of our method against WTFPAD, the typical and applicable defense in practical scenarios.

## C  DATASET

**Table 8: Details for dataset from different distributions.**

| Dataset | Websites | | Time | TBB Version |
|---|---|---|---|---|
| | Monitored | Unmonitored | | |
| Wang | Wang100 | Wang9000 | 2013 | 3.X |
| AWF | AWF100 AWF775 | AWF9000 | 2016 | 6.X |
| DF | DF95 | DF9000 | 2016 | 6.X |
| CF | CF500 | CF9000 | 2023 | 12.X |

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
