# OpenReview forum: "Contrastive Fingerprinting: A Novel Website Fingerprinting Attack over Few-shot Traces"
_ACM.org/TheWebConf/2024/Conference — TheWebConf24_

### Official Review · Reviewer_Wr7p · 2023-11-17

**Novelty:** 3
**Technical Quality:** 5

**Review:**

Summary
=======
This paper discusses the weaknesses of existing website fingerprint attacks: (i) time to develop a classifier and (ii) adaptability to new distributions. Based on these weaknesses, the authors apply contrastive learning to build the classifier and show that the new approach is effective for both open-world and close-world scenarios.

Strengths
=======
+ The discussions about existing techniques and their weaknesses are nicely written.
+ Applying contrastive learning to the problem can address the limitations of existing approaches.
+ The new attack can motivate further development of defenses.

Weaknesses
=========
- While the use of contrastive learning seems effective, the technical challenges are not articulated.

**Questions:**

1. For an open-world scenario with websites not being monitored, how to adjust the linear classifier to accommodate those websites?
2. The paper criticizes triplet loss to be inefficient. This makes the reviewer curious about the efficiency of the proposed approach. Could the authors comment on it, especially the advantage over triplet loss approaches?

**Reviewer Confidence:**

1: The reviewer's evaluation is an educated guess

**Scope:**

4: The work is relevant to the Web and to the track, and is of broad interest to the community

---

### Official Review · Reviewer_FrJk · 2023-11-27

**Novelty:** 5
**Technical Quality:** 2

**Review:**

This paper proposes a new method for training deep neural networks to perform website fingerprinting attacks, where the task is to identify which websites a user visits from the corresponding encrypted web traces. The proposed method leverages a contrastive objective, whereby the network learns representations that map traces from the same site to be close together and traces from different sites to be far apart.

The proposed method is reasonable, and the results are convincingly better than baseline methods. I'm not an expert in website fingerprinting and am thus not sure whether the baselines are appropriately conducted or comparisons are fair.

My main comments/concerns are with the writing of the paper. For me, the paper was _extremely_ hard to follow. In fact, it's possible that I have fundamentally misunderstood something since I had so much trouble reading the paper despite spending hours on it. I would highly recommend the authors take a few steps to address this:
  - The paper is littered with grammatical errors that should be carefully proofread---sentences like "We will discuss each part of the contrastive network for details in Section3.2," for example, are easily caught with a modern spell checker and significantly hinder reading of the paper.
  - Similarly, many of the transition words do not make sense (see, e.g., L202-209).
  - There are several sentences that are currently imprecise or unsubstantiated and in need of citations---for just one example, "However, in TF, an anchor sample only considers a single positive and negative sample at a time, which significantly reduces the convergence speed and effectiveness of the model." - there are several instances of this throughout the paper that should be fixed.

**Questions:**

- Why is the open-world evaluation not done using any baselines? Are they not applicable here?
- Is the main idea behind the success of this method that other methods are not able to leverage unsupervised data? If not, how are AF and TF leveraging unlabeled data? If so, how do you control for the amount of unlabeled data used?

**Reviewer Confidence:**

2: The reviewer is willing to defend the evaluation, but it is likely that the reviewer did not understand parts of the paper

**Scope:**

3: The work is somewhat relevant to the Web and to the track, and is of narrow interest to a sub-community

---

### Official Review · Reviewer_tZvE · 2023-11-28

**Novelty:** 5
**Technical Quality:** 5

**Review:**

This paper introduced a high-performance Website Fingerprinting (WF) attack called Contrastive Fingerprinting (CF). CF attack utilizes contrastive learning and data augmentation to achieve high accuracy on few training data. The result indicates that CF achieved high accuracy on most scenarios even when datasets are complicated.

This work proposed a website attack with high accuracy which is very significant for detecting threats accurately and efficiently. The concept of this paper on Contrastive Fingerprinting is novel.

Strength:
1. The paper effectively addresses the challenge of website fingerprinting with few training samples, which is a significant improvement over traditional methods that require extensive data.
2. Since a few number of training sets are needed, CF is more practical for use in real-world networks compared to traditional supervised learning methods
3. By training with augmented traces, the framework enhances its robustness to dynamic network conditions and varying trace distributions.

Weakness:
1. The Feature Extractor in the Contrastive network may be computational expensive and complex to implement.
2. Though CF shows an impressive performance, it's still challenging to apply this method on real-world applications because the real-world scenarios will vary a lot.
3. This work may raises ethical concerns. To obtain better results, it's inevitable to face with problems like user privacy leaking or malicious usage of website attaching.

**Questions:**

Could you elaborate on more details about the computational complexity and resource requirements for implementing the Contrastive Fingerprinting (CF) method? Especially the Feature Extractor?

**Ethics Review Description:**

To obtain better results, it might face with problems like user privacy leaking or malicious usage of website attaching.

**Reviewer Confidence:**

3: The reviewer is confident but not certain that the evaluation is correct

**Scope:**

4: The work is relevant to the Web and to the track, and is of broad interest to the community

---

### Official Review · Reviewer_W5jf · 2023-11-28

**Novelty:** 5
**Technical Quality:** 5

**Review:**

The authors present a high-accuracy website fingerprinting attack, which they refer to as Contrastive Fingerprinting , which leverages contrastive learning and data augmentation. They perform extensive experimental evaluation and outperform existing baselines. The appendix should appear after the references.

**Questions:**

NA

**Reviewer Confidence:**

1: The reviewer's evaluation is an educated guess

**Scope:**

3: The work is somewhat relevant to the Web and to the track, and is of narrow interest to a sub-community

---

### Official Review · Reviewer_S8pD · 2023-12-01

**Novelty:** 5
**Technical Quality:** 5

**Review:**

The paper presents a novel approach to website fingerprinting (WF) attacks, termed Contrastive Fingerprinting (CF). This method leverages contrastive learning and data augmentation to enhance WF attacks in scenarios with limited training data (few-shot scenarios). It the limitations of traditional deep learning-based WF attacks, which require extensive training data and suffer from long bootstrap times and sensitivity to changes in data distribution. CF employs a two-stage process involving pre-training and few-shot training. The pre-training stage involves training a contrastive network using a dataset with abundant labeled data. This process produces a feature extractor that is then utilized in the few-shot training stage. The method's efficacy is evaluated using extensive experiments on various datasets, considering different Tor Browser Bundle (TBB) versions and network conditions.

## Strenghts
- Novel approach using contrastive learning in the context of WF attacks; significant advancement over traditional methods
- Evaluation demonstrates strong performance in scenarios with different distributions of website traces

## Areas of Improvements
- More experiments in closed world than in open world evaluation. Given that open world evaluation is more realistic - it might be good to perform all studies with open world setup.
- Evaluation not performed with Adaptive defenses

**Questions:**

- How does CF's robustness depend on different network conditions?
- What are the potential directions for future work to enhance CF, particularly in terms of scalability and adaptability to new types of web traffic or encryption protocols?

**Reviewer Confidence:**

2: The reviewer is willing to defend the evaluation, but it is likely that the reviewer did not understand parts of the paper

**Scope:**

4: The work is relevant to the Web and to the track, and is of broad interest to the community

---

### Decision · Program_Chairs · 2024-01-22

**Decision:**

Accept

**Comment:**

We support the area chair's recommendation (below) to accept this paper. We ask the authors to add a discussion about the ethics of conducting this research in the Discussion section (A) for camera-ready submission. The authors already engage with this topic in the rebuttal stage. It is important for the readers to be aware of the nuances of ethics of research in this space.

"The paper presents a new type of website fingerprinting attack based on contrastive learning and data augmentation. The attacks are aimed for the few-shot training scenario. The reviewers found the idea to be interesting and the experimental results (including the new ones reported in the rebuttal) to be promising."